# The Beneficial Effect of Personalized Lifestyle Intervention in Chronic Kidney Disease Follow-Up Project for National Health Insurance Specific Health Checkup: A Five-Year Community-Based Cohort Study

**DOI:** 10.3390/medicina58111529

**Published:** 2022-10-26

**Authors:** Hidemi Takeuchi, Haruhito A. Uchida, Katsuyoshi Katayama, Natsumi Matsuoka-Uchiyama, Shugo Okamoto, Yasuhiro Onishi, Yuka Okuyama, Ryoko Umebayashi, Kodai Miyaji, Akiko Kai, Izumi Matsumoto, Keiko Taniguchi, Fukiko Yamashita, Tsutomu Emi, Hitoshi Sugiyama, Jun Wada

**Affiliations:** 1Department of Nephrology, Rheumatology, Endocrinology and Metabolism, Okayama University Faculty of Medicine, Dentistry, and Pharmaceutical Science, 2-5-1 Shikata-cho, Kita-ku, Okayama 700-8558, Japan; 2Department of Chronic Kidney Disease and Cardiovascular Disease, Okayama University Faculty of Medicine, Dentistry, and Pharmaceutical Science, Okayama 700-8558, Japan; 3Department of Health and Welfare, Division of Health Promotion, Mimasaka City Health Center, Mimasaka 707-0014, Japan; 4Kawasaki Medical School General Medical Center, Okayama 700-8505, Japan; 5Kawasaki College of Allied Health Professions, Okayama 700-0821, Japan

**Keywords:** chronic kidney disease, specific medical health check-up, home-visit type lifestyle intervention, CKD exacerbation

## Abstract

*Background and Objectives:* Mimasaka city is a relatively small city with a population of 28,381, and an aging rate (≥65 years old) of 38.9%, where only one nephrology clinic is available. Since 2013, the city has conducted its own unique lifestyle intervention for the participants of the National Health Insurance specific medical health checkup, aiming to prevent the progression of chronic kidney disease (CKD) severity. *Materials and Methods:* The persons in National Health Insurance specific medical health checkup (40–74 years old) conducted in Mimasaka city in 2013, with eGFR less than 50 mL/min/1.73 m² or 50–90 mL/min/1.73 m² with urine dipstick protein 1+ or more, were registered for the CKD follow-up project, as high-risk subjects for advanced renal dysfunction. Municipal workers directly visited the subjects’ homes to provide individual health guidance and encourage medical consultation. We aimed to examine the effect of home-visit intervention on the changes of renal function and related factors until 2017. *Results:* The number of the high-risk subjects who continuously received the health checkup until 2017 was 63, and only 23 (36.5%) visited a medical institution in the first year. The eGFR decreased by only 0.4 mL/min/1.73 m²/year, and the subjects with urinary protein 1+ or higher decreased significantly from 20 (31.7%) to 9 (14.3%) (*p* = 0.034) in the high-risk subjects. The changes in eGFR and urinary protein was almost in the same fashion regardless of their medical institution visits. Next, we examined the effects of various factors on ΔeGFR, the changes of eGFR from 2013 to 2017, by multivariate linear regression analysis. The effects of medical institution visit were not significant, and the degree of urinary protein (coefficient B: 4.503, β: 0.705, *p* < 0.001), age (coefficient B: 4.753, β: 0.341, *p* = 0.004), and smoking (coefficient B: 5.878, β: 0.295, *p* = 0.031) had independent significant effects, indicating that they were the factors exacerbating the decrease in eGFR from the baseline. *Conclusions:* The personalized lifestyle intervention by home-visit in CKD follow-up project showed the possibility of beneficial effects on the deterioration of renal function. This may be an efficient method to change behavior in a small community with limited medical resources.

## 1. Introduction

Chronic kidney disease (CKD) is a worldwide health problem, which has a high global prevalence estimated between 11 and 13% [1], and it is estimated that 10–12% of Japanese adults (over 10 million people) have CKD [2,3]. Furthermore, CKD is an established risk factor for end-stage renal disease (ESRD), cardiovascular disease (CVD), and all-cause mortality [4,5]. CKD is a lifestyle disease and aging is related to the progression of CKD; thus, it is a concern that the number of patients with CKD will continue to increase in an aging society. The large health care cost of dialysis for ESRD is also an important issue. Therefore, the prevention of CKD progression is one of the major public health concerns. The basic method of inhibiting the progression of CKD is multidisciplinary treatment including medications for hypertension, diabetes, hyperlipidemia, anemia, and hyperuricemia. Life-style modifications including diet therapy centered on salt reduction, exercise, and smoking cessation are also known to be effective at slowing the development of CKD. However, the effect of lifestyle guidance on the development of CKD is unclear.

A Japanese specific health checkup system which targets the National Health Insurance population aged 40 to 74 years old, is carried out for the purpose of prevention and early detection of lifestyle-related diseases, such as hypertension, dyslipidemia, diabetes, and metabolic syndrome, recently also with a focus on CKD.

According to the world population prospects, aging and depopulation have emerged as two important issues in many developed countries. As in other developed countries, aging and small size community are increasing in Japan [6]. Mimasaka city is a small area with a population of 28,381, an aging rate is 38.9% (one in 2.6 people aged 65 or over), and with limited medical institutions, as there are only 17 internal medicine clinics and only 1 nephrology clinic. Since 2013, Mimasaka city has provided generous and thorough lifestyle guidance, and recommendations of medical consultation to groups with high risk of CKD progression as a CKD follow-up project in the National Health Insurance specific medical health checkup. The approach was a home-visit type lifestyle intervention which is unique to the small community. Since the intervention of medical treatment by a nephrologist in this city was difficult due to limited medical institutions, we supposed that the effect of life guidance by home-visit would be significant. Indeed, while the effectiveness of institutional group education programs has been shown [7,8,9,10], the effectiveness of personalized interventions such as home-visit guidance remains unclear. Here, we followed up to the 5th year of the CKD follow-up project in Mimasaka city and verified the effect of home-visit type lifestyle intervention on renal function.

## 2. Materials and Methods

### 2.1. Study Subjects and Intervention

This study was a longitudinal and observational investigation. The participants were 2376 community residents aged 40 to 74 years old who received a national health insurance specific health checkup in Mimasaka city in 2013. Generally, participants who were diagnosed with metabolic syndrome [11], and the following risk holders who had not been treated at medical institutions become subjects to specific health guidance. The definition of each risk was as follows: (1) hyperglycemia: fasting plasma glucose levels ≥ 100 mg/dL or HbA1c ≥ 5.2%, (2) hypertension: systolic blood pressure ≥ 130 mmHg or diastolic blood pressure ≥ 85 mmHg, (3) dyslipidemia: serum triglycerides concentration 150 mg/dL or HDL-cholesterol concentration < 40 mg/dL, (4) hyperuricemia: serum uric acid concentration ≥ 7.0 mg/dL, in accordance with the health guidance program such as sending notification of one’s own health condition and motivating to take a spontaneous action for health that the Ministry of Health, Labor, and Welfare in Japan advocated in 2007 [12]. A notification of the recommendation for receiving health guidance was sent to each of the risk groups.

In addition to the specific health guidance, Mimasaka city has established its own CKD follow-up criterion of the high-risk subjects for advanced renal dysfunction, and provided home-visit lifestyle guidance, recommended medical institution visits and, after visiting the home, continued to provide lifestyle guidance. The CKD follow-up criterion for the high-risk subjects is as follows; estimated glomerular filtration rate (eGFR) < 50 mL/min/1.73 m^2^ or urinary protein positive or higher (including eGFR ≥ 50) in urinalysis. The participants who met the criteria were recruited as the high-risk subjects of this project.

After recruitment, we provided the lifestyle guidance by visiting the high-risk subjects’ homes, as follows. We used the instructional booklet made with reference to the dietary and lifestyle guidance manuals, which the Japanese Society of Nephrology produced based on the FROM-J study [10]. Dietary salt intake was recommended less than 6 g/day, and protein intake was restricted according to the CKD stage (0.8–1.0 g/kg/day for stage 3a, 0.6–0.8 g/kg/day for stage 3b, 4, and 5). In addition, we provided subjects with an electronic salinity meter and instructed them to use it when cooking soup. If the taste was not enough, we recommend using dashi and spices. As for lifestyle guidance, body weight target setting with BMI less than 25 kg/ m^2^, exercise, smoking cessation, and drinking restriction guidance were given dependent of the condition of the subjects. Emphasizing communication and coaching, we listened to the participants’ living circumstances and then suggested the actual proposals that the participants could easily implement in small steps and realistically. For evaluation of effects and self-monitoring, we asked them to record their blood pressure, weight, meal content, and behavior in the notebook, and we continued to give feedback based on the memo of the notebook by visiting their homes or by telephone.

We enrolled the high-risk subjects for advanced renal dysfunction who received a health checkup in 2013 in this study. We investigated the data of health checkups between 2013 and 2017 and examined the efficacy of this project on renal function. Local governments sent an official letter to residents to receive an annual health checkup of their own volition.

### 2.2. Baseline Measurements

HbA1c, high-density lipoprotein cholesterol (HDL-C), low-density lipoprotein cholesterol (LDL-C), triglyceride levels, uric acid, aspartate aminotransferase (AST), alanine aminotransferase (ALT), γ-glutamyl transpeptidase (γ-GTP), and creatinine were measured. All blood and urine analyses were performed at the local laboratories. A urinalysis was performed with the dipstick measurement of a single spot urine specimen collected at the health checkup. Dipstick urinalysis was performed manually, and results were recorded as ±, 1+, 2+, and 3+ [13]. The Japanese Committee for Clinical Laboratory Standards (Available online: http://jccls.org/ (accessed on 21 November 2021)) recommends that all urine dipstick results of 1+ correspond to a urinary protein level of 30 mg/dL. Thus, in this study, proteinuria was defined as 1+ or more by dipstick test. eGFR was calculated using the eGFR formula for Japanese (defined by the Japanese Society of Nephrology) [14]. CKD was defined as a reduced eGFR (<60 mL/min/1.73 m^2^) or the presence of proteinuria defined as 1+ or more according to the previous report [15]. The Japanese Society of Nephrology set the standards of referral to a nephrologist (eGFR < 50 mL/min/1.73 m^2^: people who aged between 40 and 70) [14]. Therefore, the authors defined the subjects who had eGFR < 50 mL/min/1.73 m^2^ in this study as CKD follow-up project target. The renal status of participants was evaluated using CKD heat map [16]. In Japan, measurement of urine albumin concentration is covered by health insurance only for patients with early stage diabetic nephropathy. Therefore, dipstick urinalysis is common for health checkups. To adapt the result of dipstick urinalysis to CKD heat map, we defined as A1, ± and 1+ as A2, and 2+ or more as A3 [13,17]. The amount of decrease in eGFR from 2013 to 2017 was defined as ΔeGFR.

### 2.3. Statistics

All data are presented as the mean ± standard deviation unless otherwise noted. Differences were analyzed by Student’s *t*-test or Chi-square test where appropriate. Multivariate linear regression analysis determines ΔeGFR as an outcome, and age, gender, smoking, drinking habits, medical institution visit, history of heart disease, history of stroke, hypertension, dyslipidemia, diabetes, hyperuricemia, baseline eGFR, and the degree of urine protein were used as explanatory variables. A difference of *p* < 0.05 was taken as statistically significant. All data were analyzed using Sigma Plot for Windows (version 14.0, Systat Software Inc., San Jose, CA, USA).

## 3. Results

### 3.1. Study Participants

As shown in Table 1, the number of participants who received the Mimasaka city National Health Insurance specific medical health checkup in 2013 was 2376, consisted of 1102 males and 1274 females (male/female ratio 0.46), and the mean age was 65.0 ± 7.2 years (40–75 years), mean serum creatinine 0.74 ± 0.29 mg/dL, mean eGFR 75.2 ± 15.4 mL/min/1.73 m^2^, and participants with urinary protein ± or higher were 126 (5.3%). There were 414 participants (17.4%) who corresponded to CKD. As shown in Figure 1, of the 2376 participants, 299 corresponded to the subjects of specific health guidance (90 participants were equivalent to metabolic syndrome and 209 participants were equivalent to pre-metabolic syndrome), and there were 118 participants (5.0%) in the high-risk group for advanced renal dysfunction. There were only 12 duplicate subjects between high-risk subjects and specific health guidance subjects. This indicates that the remaining 106 subjects would have been overlooked if this project had not been conducted. During the CKD follow-up project from 2013 to 2017, as displayed in Figure 2, 63 subjects completed the entire course. 

The results of the specific medical examination conducted in 2013 are shown. Specific health guidance and high-risk subjects for advanced renal dysfunction are specified in different ways, and some participants overlap. CKD, chronic kidney disease; UP, urinary protein.

### 3.2. The Parameter Changes in High-Risk Subjects for Advanced Renal Dysfunction

Comparing the various parameters of 63 subjects, who accomplished the follow-up to 2017, between 2013 and 2017 (Table 2), the significant changes were confirmed in the serum levels of HDL-C, creatinine, and urinary protein by paired *t*-test or Chi-square test. Regarding the lipid profile, HDL-C was slightly worsened with a significant difference, however LDL-C was improved although there was no significant difference. There was no significant difference in uric acid, but there was a tendency for improvement. There was almost no change in blood pressure, BMI, or HbA1c. Regarding renal function, creatinine significantly deteriorated from 1.08 mg/dL to 1.16 mg/dL; however, when converted to eGFR, there was no significant difference with physiological aging. In addition, the subjects with urinary protein 1+ or higher decreased significantly from 20 (31.7%) to 9 (14.3%) (*p* = 0.034). Figure 3 shows the annual transition of renal function, and the eGFR decreased by only 1.6 mL/min/1.73 m^2^ in 4 years, which means that eGFR decreased by 0.4 mL/min/1.73 m^2^/year. This may be due to the improvement of proteinuria.

When converted to the CKD heatmap, green groups that were not observed in 2013, appeared from 2014, as shown in Figure 4. The number of red groups increased slightly; however, the number of oranges decreased and the number of yellows increased.

### 3.3. The Effect of Medical Institution Visit on Renal Function

Of the 63 subjects, only 23 subjects visited a medical institution after home-visit guidance by a public health nurse and continued to receive treatment. However, the remaining 40 participants could not visit a medical institution. To examine the effect of medical consultation on renal function, the annual transition of renal function was evaluated by the presence or absence of medical institution visit (Figure 5). The eGFR of the subjects who visited the clinic decreased from 47.0 to 45.2 mL/min/1.73 m^2^, thus the ΔeGFR was 1.8 mL/min/1.73 m^2^. On the other hand, the eGFR of non-visited subjects decreased from 52.9 to 51.3 mL/min/1.73 m^2^, thus the ΔeGFR resulted in 1.6 mL/min/1.73 m^2^. The difference in ΔGFR between the groups was small and not statistically significant. In addition, the urinary protein was improved to the same extent regardless of whether or not there were medical institution visits. We examined the effects of various factors on ΔeGFR from 2013 to 2017 by multivariate linear regression analysis, and the effects of medical institution visit were not significant, and the degree of urinary protein (coefficient B: 4.503, β: 0.705, *p* < 0.001), age (coefficient B: 4.753, β: 0.341, *p* = 0.004), and smoking (coefficient B: 5.878, β: 0.295, *p* = 0.031) had independent significant effects. On the other hand, the higher the baseline eGFR, the better the renal prognosis (coefficient B: −1.979, β: −0.383, *p* = 0.009) (Table 3). The above results indicate that eGFR is more likely to decrease in subjects with older age, higher urine protein content, and smoking habits. Conversely, participants with high baseline eGFR are likely to preserve their eGFR.

The bar graph on the left is the evaluation of urinary protein of subjects who visited a medical institution, and the bar graph on the right is the evaluation of urinary protein of subjects who could not visit a medical institution. The blue line graph is the transition of eGFR of the subjects who visited the medical institution, and the orange line graph is the transition of the eGFR of the subjects who could not visit the medical institution.

## 4. Discussion

This study showed that home-visit type lifestyle intervention in CKD follow-up project had a possibility of beneficial effect on renal function in participants with high risk of CKD progression in the National Health Insurance specific medical health checkup. Although our study was an observational study with several unmeasurable confounding factors and without the control groups to compare with the intervention group, the results demonstrated the effective non-drug CKD intervention method by public health nurses as personalized lifestyle guidance for patients with CKD.

Although many of the subjects in this study were unable to receive medical therapy by a nephrologist, they had only 1.6 mL/min/1.73 m^2^ reduction of eGFR in 4 years. Most beneficial effects of intervention by this project resulted from unique feature of face-to-face, home-visit style, and continuous health guidance supported by public health nurses and nutritionists belonging to the city health center. Face-to-face guidance can provide better understanding of lifestyle modification “how to do”, because subjects can ask any kind of question, then receive supportive information immediately. Mailing is a common method to inform subjects of the risk of CKD, but this is a kind of one-way notification, thus subjects need spontaneous action to solve any of their questions by themselves. It is true that multicomponent educational interventions were also effective in CKD patient care [7,8,9,10], however, in many cases, group education programs have been conducted at a single time, In addition, these were not individualized. From the perspective of individualization, this project might be more effective due to the personalized intervention that focused on individual issues. We also supposed that the continuous contact with the subjects would sustainably affect participants’ lifestyle behavior, including promoting reduction of salt intake, drinking water, active exercise, and avoiding nephrotoxic drugs such as NSAIDs. We are confident that the continuation of renoprotective lifestyle modifications are major contributors for preventing the advanced renal dysfunction in this project. 

There are data of the CKD-JAC study performed to confirm the effect of multidisciplinary treatment by nephrologists in Japan [18]. The mean 3.9-year follow-up results showed that the rate of decline in eGFR was 1.93 mL/min/1.73 m^2^/year in CKD stage G3a and 2.06 mL/min/1.73 m^2^/year in CKD stage G3b [18]. In our study, eGFR decreased by only 0.4 mL/min/1.73 m^2^ /year, despite the lack of medical care for most participants. In CKD-JAC study, in addition to drug treatment by nephrologists, probably non-drug treatment including nutritional and lifestyle guidance were provided at the hospital; however, the frequency and sustainability were unknown and they could be irregular. Irregular interventions may interrupt lifestyle modification and lose the effects of health guidance to some extent. In contrast, in our project, the public health nurses regularly contacted participants by telephone. Therefore, it seems that there were few cases in which lifestyle modification was interrupted due to our regular intervention. The FROM-J study also exists as a comparison of the efficacy of the present study, using more multidisciplinary medical by general physicians, lifestyle, and nutritional therapy interventions in Japan [10]. In the FROM-J study, all patients were managed in accordance with the national current CKD guidelines, and the intervention group received additional interventions: patients received both educational intervention for lifestyle modification and a CKD status letter, attempting to prevent their withdrawal from treatment, and the general physicians received data sheets to facilitate reducing the gap between target and practice. The 3.5-year follow-up results showed the average eGFR deterioration rate tended to be lower in intervention group (conventional group: 2.6 ± 5.8 mL/min/1.73 m^2^/year, intervention group: 2.4 ± 5.1 mL/min/1.73 m^2^/year, *p* = 0.07), and a significant difference in eGFR deterioration rate was observed in subjects with Stage 3 CKD (conventional group: 2.4 ± 5.9 mL/min/1.73 m^2^/year, intervention group: 1.9 ± 4.4 mL/min/1.73 m^2^/year, *p* = 0.03) [10]. Although the FROM-J study clearly showed the beneficial effect of lifestyle modification, our data were still better. While FROM-J was an institution-visit type life guidance, our project was a home-visit type life guidance. Therefore, the participation rate of life guidance was 100% in our study, although the participation rate in FROM-J study was unknown. It seems that the strength of this project is that the lifestyle guidance can be sustained without fail. Furthermore, although in a follow-up study of Japanese in CKD stage 1–2 [19], the average annual decrease in eGFR was 1.64 mL/min/1.73 m^2^/year, the average annual decrease in eGFR in our study was far below, suggesting that the intervention in our project was more effective. In a double-blind, randomized controlled trial evaluating the effects of salt restriction in patients with CKD, salt restriction reduced extracellular fluid volume, reduced blood pressure, and halved urinary protein and albumin [20]. In addition, a review examining the relationship between dietary patterns and renal function reported that dietary patterns with low dietary acid load may delay the progression of CKD [21]. In this project, dietary guidance was provided with reference to the Chronic Kidney Disease Lifestyle and Dietary Guidance Manual for Doctors and Comedies prepared by the Japanese Society of Nephrology, and participants were also instructed on salt reduction and protein intake. In addition, by providing a salt measuring instrument, it was possible for subjects to measure the salt content in the diet at home. This intervention did not significantly lower blood pressure, but it significantly reduced the number of subjects with urinary protein + or higher. The same changes were observed regardless of their medical institution visit; thus, we assumed that this might be the effect of continuous dietary guidance.

This is an intervention that might be difficult to adapt large-population areas because these areas have too many individuals with CKD. However, in a small community, public health nurses and examiners participating in medical health checkup are intimate due to the small number of population and the small size of the community, which appears a very effective method. In addition, it is a non-drug treatment that can be performed even in areas where there are few medical institutions, and the medical cost is inexpensive. Thus, it is considered to be a preferable model method that will be effective in small-population areas with limited medical institutions. Although CKD awareness and treatment that take advantage of regional characteristics are required, this project suggests that an effective model of an efficient intervention that can be carried out in small-population areas, which will increase in many developed countries.

When assessing renal function in different races and ethnicities, eGFR based on creatinine sometimes has various differences depending on physique and genetic predisposition, resulting in underestimation or overestimation. Recently, the National Kidney Foundation (NKF) and the American Society of Nephrology (ASN) established a Task Force and developed new eGFR equations for the population in the United States [22]. The new eGFR is called the CKD-EPI 2021 eGFR creatinine equation, which is a race-free estimation formula recommended by the NKF-ASN Task Force. In regions such as the United States where various races are mixed, CKD-EPI 2021 eGFR creatinine equation reduces the influence of eGFR due to race; however, it is not yet sufficiently validated in many countries and for many races, therefore, it is unclear what kind of differences will occur by races and ethnicities. The CKD-EPI 2021 eGFR using creatinine and cystatin C (eGFRcr-cys) is reported to be more accurate regardless of race [23]. When deciding targets for CKD life guidance in different races and ethnicities, the CKD-EPI 2021 eGFRcr-cys would be useful in mixed-race areas and in countries where eGFR calculations adapted to their own country have not been established.

This study has several limitations. First, this study was based on a single arm and observational data and the authors could not set the control group such as the people who did not participate in the health checkup or who did not receive the project intervention. Thus, the real effect of our intervention might be under- or overestimated. Second, there was a selection bias. The number of subjects eligible for the health checkup in Mimasaka city aged between 40 and 74 was 28,381 in 2013. Basically, Japan has a universal health insurance coverage. This health checkup was performed for the people who hold national health insurance. However, only 2376 subjects received a health check-up in 2013 and the number of subjects in the CKD follow-up project decreased year by year, and finally subjects were only 63. In addition, most of the remaining 63 subjects received the health checkup every time. Therefore, it appears that many subjects might be originally highly motivated as regards their health. It seems that they might change behaviors by themselves to improve their renal function. Third, the findings may have been brought by the regression toward the mean. In health checkups, the person who had the worse result naturally tended to improve in next time, the person who had the better result tended to remain. The authors might observe a part of natural drift of laboratory tests during the follow-up period. Fourth, we were unable to obtain information regarding the cause of CKD, drugs, comorbidities, frequency of exercise, nutrition habits, and amount of salt intake because these are not included in the elements of the specific health check-up system. Fifth, this method requires manpower and time to visit each person, which is not feasible in a large community such as a large-population city. Sixth, there is regional uncertainty. Since the effect depends on the regional culture, differentiation, and social conditions where the intervention is implemented, it is unclear whether the same effect as this study could be obtained in every region or for many races and ethnicities. Furthermore, the face-to-face method is difficult to adapt in situations under the COVID-19 pandemic. In such a situation, it is necessary to change to alternative methods, such as an online interview. Finally, this is a short-term observational study, and a five-year drawback is not enough, thus additional long-term intervention and follow-up are required.

## 5. Conclusions

In conclusion, a personalized lifestyle intervention by home-visit showed the possibility of beneficial effects on the deterioration of renal function in subjects with high-risk of CKD progression in a small-population area with limited medical resources.

## Figures and Tables

**Figure 1 medicina-58-01529-f001:**
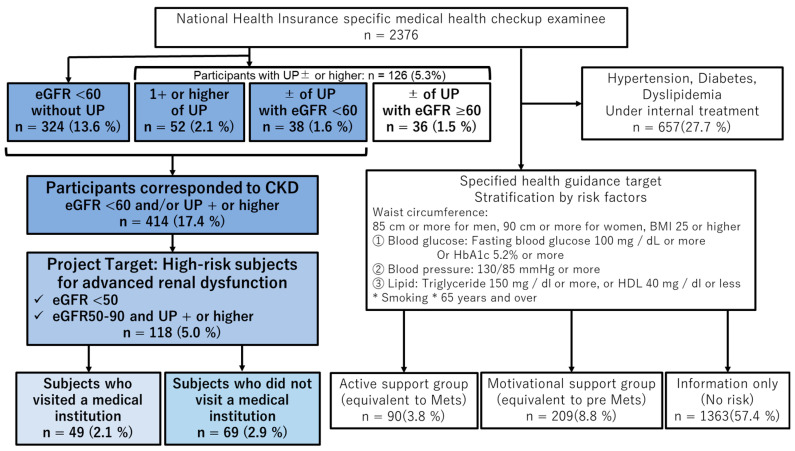
Diagram of participants enrolled in this study.

**Figure 2 medicina-58-01529-f002:**
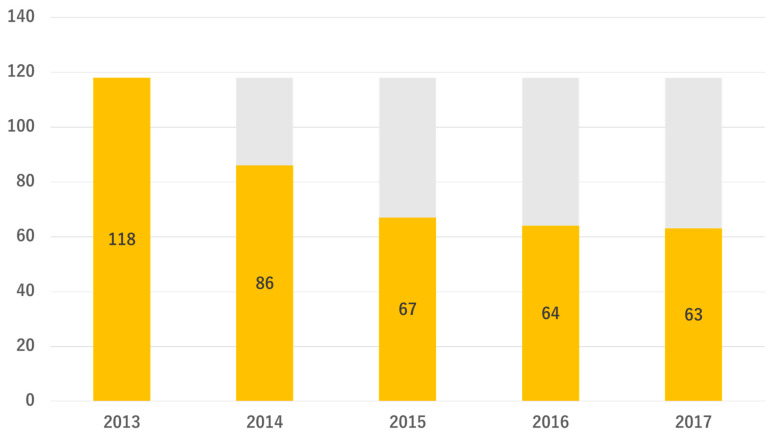
Changes in the number of high-risk subjects for advanced renal dysfunction. The orange bar shows the number of high-risk subjects in each year.

**Figure 3 medicina-58-01529-f003:**
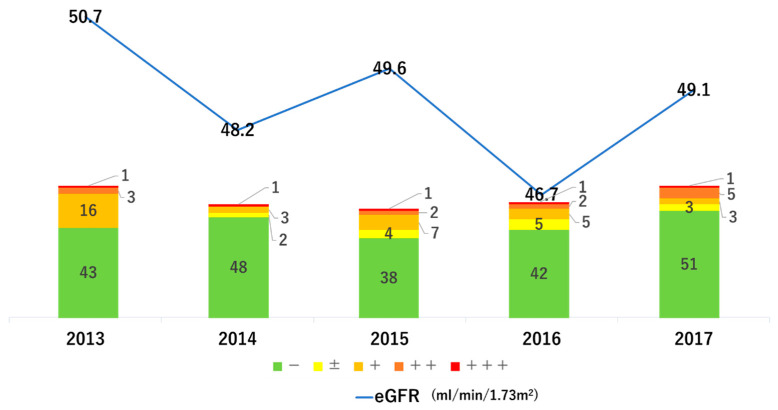
Transition of renal function and urinary protein. The bar graph is the evaluation of urinary protein of subjects, and the line graph is the transition of mean eGFR of subjects.

**Figure 4 medicina-58-01529-f004:**
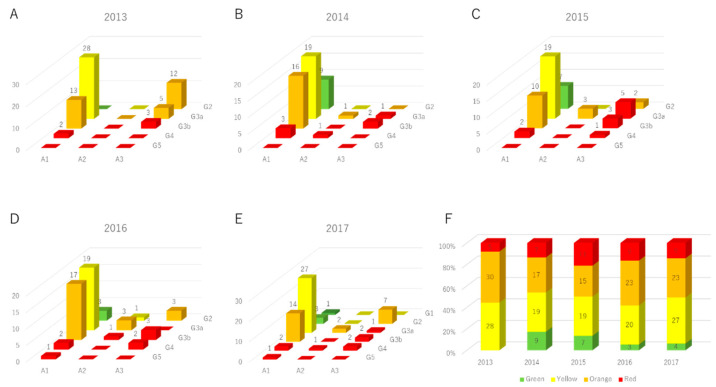
Transition of CKD heat map. (**A**) The distribution of CKD heatmaps of subjects in 2013. (**B**) The distribution of CKD heatmaps of subjects in 2014. (**C**) The distribution of CKD heatmaps of subjects in 2015. (**D**) The distribution of CKD heatmaps of subjects in 2016. (**E**) The distribution of CKD heatmaps of subjects in 2017. (**F**) The changes in the proportion of heat maps for each year.

**Figure 5 medicina-58-01529-f005:**
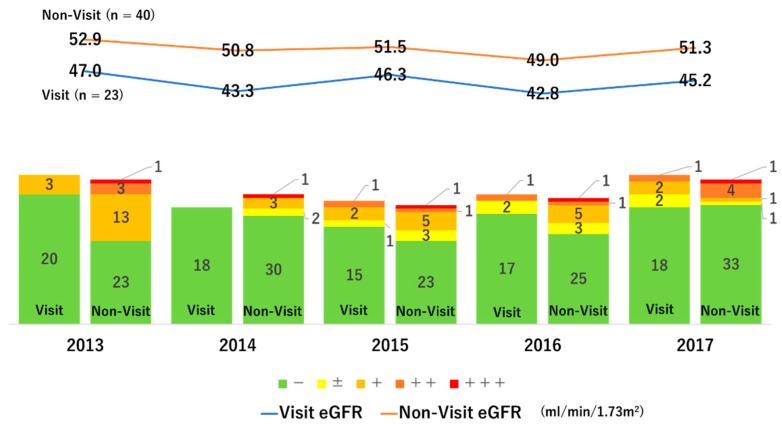
Differences in renal function and urinary protein depending on the presence or absence of medical institution visit.

**Table 1 medicina-58-01529-t001:** Baseline characteristics of participants.

Gender	Male 1102/Female 1274 (Ratio: 0.46)
Age, (years)	65.0 ± 7.2
40–49 years old, n (%)	117 (4.9%)
50–59 years old, n (%)	244 (10.3 %)
60–69 years old, n (%)	1255 (52.8 %)
70–74 years old, n (%)	760 (32.0 %)
Urine protein	
No data, n (%)	1
−, n (%)	2249 (94.7 %)
±, n (%)	74 (3.1 %)
+, n (%)	34 (1.4 %)
2+, n (%)	15 (0.6 %)
3+, n (%)	3 (0.1 %)
Renal function	
Creatinine, (mg/dL)	0.74 ± 0.29
eGFR, (mL/min/1.73 m^2^)	75.2 ± 15.4

The data are presented as the mean value ± standard deviation or n (%) of participants.

**Table 2 medicina-58-01529-t002:** Characteristics of high-risk subjects for advanced renal dysfunction in each year.

Variable	2013 (n = 63)	2014 (n = 54)	2015 (n = 52)	2016 (n = 55)	2017 (n = 63)	*p* Value
Age, (years)	67.2 ± 5.0					
Gender (male), n (%)	36 (56.3 %)					
Systolic blood pressure	133 ± 16	131 ± 16	130 ± 16	131 ± 19	135 ± 19	0.546
Diastolic blood pressure	77 ± 11	76 ± 10	74 ± 11	77 ± 12	77 ± 11	0.657
Body mass index, (kg/m²)	23.8 ± 3.4	23.3 ± 3.0	23.6 ± 3.4	23.3 ± 3.5	23.6 ± 3.5	0.207
Uric acid, (mg/dL)	6.3 ± 1.6	6.3 ± 1.7	6.0 ± 1.4	5.9 ± 1.3	6.0 ± 1.3	0.084
Triglyceride, (mg/dL)	115 ± 59	113 ± 47	115 ± 53	113 ± 51	127 ± 82	0.171
HDL cholesterol, (mg/dL)	57 ± 15	61 ± 15	59 ± 14	58 ± 15	54 ± 15	0.033 *
LDL cholesterol, (mg/dL)	121 ± 29	122 ± 27	123 ± 26	117 ± 26	114 ± 30	0.059
AST, (IU/L)	23 ± 9	23 ± 6	23 ± 7	24 ± 9	24 ± 12	0.355
ALT, (IU/L)	18 ± 11	18 ± 7	18 ± 9	18 ± 10	18 ± 10	0.902
γ-GTP, (IU/L)	28 ± 20	29 ± 21	29 ± 23	28 ± 18	32 ± 32	0.169
HbA1c, (%)	5.8 ± 0.6	5.8 ± 0.5	5.8 ± 0.5	5.8 ± 0.6	5.8 ± 0.7	0.457
Creatinine, (mg/dL)	1.08 ± 0.32	1.14 ± 0.40	1.10 ± 0.37	1.19 ± 0.51	1.16 ± 0.56	0.047 *
eGFR, (mL/min/1.73 m²)	50.7 ± 13.6	48.2 ± 13.6	49.6 ± 12.7	46.7 ± 13.7	49.1 ± 15.7	0.068
Urine protein						
−, n (%)	43 (68.3 %)	48 (88.9 %)	38 (73.1 %)	42 (76.4 %)	51 (81.0 %)	0.034 *
±, n (%)	0 (0 %)	2 (3.7 %)	4 (7.7 %)	5 (9.1 %)	3 (4.8 %)	
+, n (%)	16 (25.4 %)	3 (5.6 %)	7 (13.5 %)	5 (9.1 %)	3 (4.8 %)	
2+, n (%)	3 (4.8 %)	0 (0 %)	2 (3.8 %)	2 (3.6 %)	5 (7.9 %)	
3+, n (%)	1 (1.6 %)	1 (1.9 %)	1 (1.9 %)	1 (1.8 %)	1 (1.6 %)	

The data are presented as the mean value ± standard deviation or n (%) of patients. *p* values are obtained by paired *t*-test or Chi-square test with comparison of 2013 and 2017 data. * *p* < 0.05.

**Table 3 medicina-58-01529-t003:** Multivariate linear regression for factors influencing ΔeGFR.

Variables	B	SE	β	t Value	*p* Value
Age (10 years)	4.753	1.574	0.341	3.019	0.004 **
Male (1/0)	−1.229	1.793	−0.087	−0.685	0.496
Current Smoker (1/0)	5.878	2.645	0.295	2.222	0.031 *
Drinking Habits (1/0)	3.522	1.937	0.210	1.818	0.075
Medical Institution Visit (1/0)	0.953	1.625	0.066	0.586	0.560
History of Heart Disease (1/0)	0.850	3.155	0.030	0.270	0.789
History of Stroke (1/0)	1.018	2.516	0.043	0.405	0.687
Hypertension (1/0)	−1.893	1.561	−0.134	−1.213	0.231
Dyslipidemia (1/0)	−0.355	1.535	−0.025	−0.231	0.818
Diabetes (1/0)	−2.754	3.009	−0.116	−0.915	0.364
Hyperuricemia (1/0)	−0.511	1.792	−0.035	−0.285	0.777
Baseline eGFR (/10 mL/min/1.73 m²)	−1.979	0.723	−0.383	−2.736	0.009 **
Urine protein (−: 0, ±: 1, 1+: 2, 2+: 3, 3+: 4)	4.503	0.987	0.705	4.560	<0.001 **

Coefficient B, β, SE, t and *p* values were obtained by a multivariate linear regression analysis. * *p* < 0.05, ** *p* < 0.01. SE, standard error.

## Data Availability

Not applicable.

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
