# Peer review of "The Beneficial Effect of Personalized Lifestyle Intervention in Chronic Kidney Disease Follow-Up Project for National Health Insurance Specific Health Checkup: A Five-Year Community-Based Cohort Study"

_medicina, 2022, doi:10.3390/medicina58111529_

Round 1

Reviewer 1 Report

Chronic kidney disease (CKD) has become an important issue for global public health and the economy. Multidisciplinary methods should be utilized to improve renal outcomes for patients with limited medical accessibility. Here are my suggestions for the manuscript:

01.  The home-visit intervention is the key element of this cohort study, please make a detailed introduction with examples in section 2.1.

02.  The general concept of CKD is defined as the irreversible renal impairment anatomically or functionally for more than 3 months in KDOGI and KDIGO guidelines. The definition of CKD in section 2.2 seemed to be different from those above, please make a detailed explanation of reference 15.

03.  In section 3.1, 414 people corresponded to CKD, 126 people had significant proteinuria via dipstick urinalysis and 118 people were at high risk of advanced renal dysfunction. Since there might be significant overlap among these groups, please describe and label them precisely in Figure 1.

04.  Multivariate linear regression for ΔeGFR seemed not to be appropriate owing to minimal improvement after the home-visit intervention. Multivariate logistic regression for dipstick urinalysis might be a better method in this study.

05.  There were wrong spellings of “Uremic acid” in Table 2 and “evaluaTable4” in Figure 3 ligands. Please make a thorough spelling and grammar recheck before the resubmission.

Author Response

We would greatly appreciate the time and effort of the reviewers in providing their constructive critiques.  We have provided a point-by-point response to the reviewer’s critiques in addition to revising the manuscript, and we wrote in red when we changed the sentences.

We hope that our responses and changes to the manuscript have adequately addressed the reviewers’ concerns.

For reviewer #1:

We would appreciate this reviewer for his/her constructive comments and important suggestions.  We here respond to all his/her comments point-by-point in order to make this revised manuscript properly.  We hope we could respond to his/her all kind and helpful suggestions.

Reviewer’s comment 1
The home-visit intervention is the key element of this cohort study, please make a detailed introduction with examples in section 2.1.

Response.  

     We would like to appreciate the reviewer for his/her important comments. As the reviewer took well, the specific method of the intervention we performed was the key element of this study.  Therefore, we added the description to section 2.1 of Materials and Methods paragraph as bellow.

e.g.

(Located in line 34-36 of page 4 and line 1-14 of page 5)

After recruitment, we provided the lifestyle guidance by visiting the high-risk subjects’ homes, as follows. We used the instructional booklet made with reference to the dietary and lifestyle guidance manuals, which the Japanese Society of Nephrology produced based on the FROM-J study [10]. Dietary salt intake was recommended less than 6 g/day, and protein intake was restricted according to the CKD stage (0.8-1.0 g/kg/day for stage 3a, 0.6-0.8 g/kg/day for stage 3b, 4 and 5). In addition, we provided subjects the electronic salinity meter and instruct them to use it when cooking soup. If the taste was not enough, we recommend using dashi and spices. As for lifestyle guidance, body weight target setting with BMI less than 25 kg/ m2, exercise, smoking cessation, and drinking restriction guidance were given dependent of the condition of the subjects. Emphasizing communication and coaching, we listened to the participants' living circumstances and then suggested the actual proposals that the participants could easily implement in small steps and realistically. For evaluation of effects and self-monitoring, we asked them to record their blood pressure, weight, meal content, and behavior in the notebook, and we continued to give feedback based on the memo of the notebook by visiting their homes or by telephone.

Reviewer’s comment 2

The general concept of CKD is defined as the irreversible renal impairment anatomically or functionally for more than 3 months in KDOGI and KDIGO guidelines. The definition of CKD in section 2.2 seemed to be different from those above, please make a detailed explanation of reference 15.

Response.  

We would thank the reviewer for his/her important comments.  We used the definition of CKD as follows, “CKD was defined as a reduced eGFR (< 60 mL/min/1.73m2) or the presence of proteinuria defined as 1+ or more.”  However, as he/she pointed out, originally CKD is defined as the irreversible renal impairment anatomically or functionally for more than 3 months, according to KDIGO 2012 Clinical Practice Guideline for the Evaluation and Management of Chronic Kidney Disease.  The guideline defined decreased renal function as eGFR <60 ml/min/1.73 m2 and abnormal proteinuria as presence of albuminuria.  However, in Japanese specific health checkup system, presence of albuminuria is substituted with urine dipstick results of 1+ or higher of proteinuria. In addition, since the specific health checkup is conducted once a year, irreversible abnormality for more than 3 months was excluded in many studies regarding health checkup as our study.  In addition, renal morphology abnormalities are excluded from the criteria, because imaging tests are not always performed in the health checkup.  Reference 15 is the study analyzed data from the Japanese specific health checkup system at the same time as our study, therefore we also referred to the definition of CKD which is described in method paragraph of reference 15.  There are also other specific health checkup studies that use the same definition “CKD is defined in examinees as eGFR less than 60 mL/min/1.73 m2 or qualitative analysis of urinary protein 1+ or greater at the specific health check-up”, as well.  Our definition is not particularly different from usual definitions. Several papers using similar definition are listed as below. Please refer to them.

Other references, e.g.

Kawazoe M, et al. J Clin Hypertens (Greenwich). 2021 Dec;23(12):2071-2077. Effect of chronic kidney disease on the association between hyperuricemia and new-onset hypertension in the general Japanese population: ISSA-CKD study.

Sofue T, et al. Clin Exp Nephrol. 2019 Aug;23(8):1031-1038. The effects of a participatory structured group educational program on the development of CKD: a population-based study.

Uchida D, et al. Kidney Blood Press Res. 2019;44(5):973-983. Lower Diastolic Blood Pressure was Associated with Higher Incidence of Chronic Kidney Disease in the General Population Only in those Using Antihypertensive Medications

Reviewer’s comment 3

In section 3.1, 414 people corresponded to CKD, 126 people had significant proteinuria via dipstick urinalysis and 118 people were at high risk of advanced renal dysfunction. Since there might be significant overlap among these groups, please describe and label them precisely in Figure 1.

Response.  

We would appreciate the reviewer for this important comment.  As he/she pointed out, there were several subjects who overlapped.  We modified Figure 1 to make it clear to understand.  Please make sure the attached modified figure 1 in the revised manuscript.

Reviewer’s comment 4

Multivariate linear regression for ΔeGFR seemed not to be appropriate owing to minimal improvement after the home-visit intervention. Multivariate logistic regression for dipstick urinalysis might be a better method in this study.

Response.  

We would appreciate the reviewer for his/her important suggestion.  As he/she pointed out, it was also very important to analyze the contributing factors to the improvement of dipstick urinalysis, then we performed an additional multivariate logistic regression.  Improvement of urinary protein was used as the objective variable (a participant who had an improvement was set to 1, and a participant who did not improve was set to 0), and we used the explanatory variables such as age, gender, history of heart disease, cerebral stroke, hypertension, diabetes, dyslipidemia, smoking habits, drinking habits, medical institution visit, baseline eGFR and baseline urinary protein level.  However, outliers of odds ratio such as 0.000 and infinity occurred frequently.  We tried several methods, such as changed the number and composition of input explanatory variables and excluded participants who were negative for proteinuria at baseline, however we could not obtain effective titer.  Probably, there were only 20 proteinuria-positive participants at baseline and 9 proteinuria-positive patients at the final follow-up, and the number of participants to be analyzed was very small, in result, overfitting seems to occur easily.  Then we gave up including multivariate logistic regression for dipstick urinalysis in the manuscript.

Reviewer’s comment 5

There were wrong spellings of “Uremic acid” in Table 2 and “evaluaTable4” in Figure 3 ligands. Please make a thorough spelling and grammar recheck before the resubmission.

Response.  

We would appreciate the reviewer for his/her notice.  As he/she pointed out, several descriptions were incorrect, and regarding Table 2, we corrected it to "Uric acid" instead of "Uremic acid".  However, we checked "evaluaTable4" in Figure 3 legends, there is no such description.  Overall, we rechecked the spelling and grammar, and then we corrected 2 sentences as bellow.

e.g.

(Located in 21 line of page 3)

Japanese special health checkup system → Japanese specific health checkup system

(Located in 12 line of page 18)

the elements of the special health check-up system → the elements of the specific health check-up system

Reviewer 2 Report

This is an interesting study about the progression of CKD in an underserved population and the effects of personalized interventions. There are several points of general interest that make this interesting. Few studies have looked at asian rates of progression of CKD and non pharmacologic interventions have been studied even less. Studies such as this one help understand the natural history of CKD progression. The authors should expand on this in the discussion with respect to other races and ethnicities, especially since the new GFR calculations exclude race and  how this might impact treatment decisions.

Secondly, this study is of interest due to the emphasis on prevention of progression. This is also its chief drawback as 5 years is not long enough to see beneficial effects of preventive and life style modification methods.

Overall, the authors have discussed the limitations of the study in the discussion and have done a good job of presenting it in context.

Author Response

We would greatly appreciate the time and effort of the reviewers in providing their constructive critiques.  We have provided a point-by-point response to the reviewer’s critiques in addition to revising the manuscript, and we wrote in red when we changed the sentences.

We hope that our responses and changes to the manuscript have adequately addressed the reviewers’ concerns.

For Reviewer #2:

First of all, we would like to appreciate the reviewer #2 for his/her constructive comments.  We here respond to all his/her comments in order to make this revised manuscript properly.  We hope we could respond to his/her all kind and helpful suggestions.

Reviewer’s comment 1

This is an interesting study about the progression of CKD in an underserved population and the effects of personalized interventions.  There are several points of general interest that make this interesting.  Few studies have looked at asian rates of progression of CKD and non-pharmacologic interventions have been studied even less.  Studies such as this one help understand the natural history of CKD progression.  The authors should expand on this in the discussion with respect to other races and ethnicities, especially since the new GFR calculations exclude race and how this might impact treatment decisions.

Response 

We would greatly appreciate his/her important comments.  As he/she indicated, we also consider that it is a very important issue whether a similar approach can be applied to other regions and races.  In addition, we also understand that it is very important to consider the evaluation using the CKD-EPI 2021 eGFR equation, which is a race-free estimation formula recommended by the NKF-ASN Task Force.

First, we believe that it is possible for small communities to conduct the same home-visit approach as this project.  The smaller the area of the home-visit intervention, the higher feasibility.  Another advantage of this method is that it can be implemented at low cost, therefore, it is considered useful even in regions that do not have sufficient budgets for medical care.  However, since the key to success is whether or not the participants are obedient to the instructions, we think that the honesty and culture of the intervention area are important.  Therefore, it is unclear whether there is a similar effect in other countries and regions, then we added the regional uncertainty to the limitation paragraph as bellow.

e.g.

(Located in the 14-18 line of page 18)

Sixth, there is the regional uncertainty. Since the effect depends on the regional culture, differentiation and social conditions where the intervention is implemented, it is unclear whether the same effect as this study could be obtained in every region or for many races and ethnicities.

Regarding the CKD-EPI 2021 eGFR creatinine equation, it is not yet sufficiently validated in many countries and for many races as for us Japanese population, therefore it is unclear what kind of differences will occur by races and ethnicities.  Depending on race and region, renal function may be underestimated or overestimated, therefore there is a possibility that the number of people targeted for lifestyle guidance will change, however the evaluation for the changes in renal function will be less affected.  The CKD-EPI 2021 eGFR using creatinine and cystatin C (eGFRcr-cys) is reported more accurate, thus, in regions where new GFRs will be applied in the future, the use of eGFRcr-cys will reduce the effects due to race and regional differences. We understand that it is also important to use the new GFRs, therefore, we added consideration on new GFR to the discussion paragraph as bellow.

    ↓

e.g.

(Located in the 11-27 line of page 17)

When assessing renal function in different races and ethnicities, eGFR based on creatinine sometimes has various differences depending on physique and genetic predisposition, resulting in underestimation or overestimation. Recently, the National Kidney Foundation (NKF) and the American Society of Nephrology (ASN) established a Task Force and developed new eGFR equations for the population in the United States[22]. The new eGFR is called the CKD-EPI 2021 eGFR creatinine equation, which is a race-free estimation formula recommended by the NKF-ASN Task Force. In regions such as the United States where various races are mixed, CKD-EPI 2021 eGFR creatinine equation reduces the influence of eGFR due to race, however it is not yet sufficiently validated in many countries and for many races, therefore, it is unclear what kind of differences will occur by races and ethnicities. The CKD-EPI 2021 eGFR using creatinine and cystatin C (eGFRcr-cys) is reported to be more accurate regardless of race [23]. When deciding targets for CKD life guidance in different races and ethnicities, the CKD-EPI 2021 eGFRcr-cys would be useful in mixed-race areas and in countries where eGFR calculations adapted to their own country have not been established.

Reviewer’s comment 2

Secondly, this study is of interest due to the emphasis on prevention of progression. This is also its chief drawback as 5 years is not long enough to see beneficial effects of preventive and life style modification methods.

Response

We would appreciate the reviewer for this important comment.  As he/she pointed out, we also consider that a five-year drawback is not enough, then we are continuing to follow up the participants of this project.  Once the long-term data is collected, we would like to report it in a medical journal in the future.

In accordance with his/her comment, we added sentences to the limitation paragraph as bellow.

e.g.

(Located in 20-22 line of page 18)

Finally, this is a short-term observational study and a five-year drawback is not enough, thus the additional long-term intervention and follow-up are required.

Round 2

Reviewer 1 Report

Dear authors,

Thanks for your detailed revision. Your clarification and supplementation improve the quality of the manuscript. However, we could not figure out the beneficial effect of personalized lifestyle intervention on renal function from the study, which might relate to limited case numbers and different definitions of chronic kidney disease.

Author Response

We would greatly appreciate the time and effort of the reviewer in providing his/her important comments.  We have provided a response to the reviewer’s comments in order to answer the reviewers’ concerns properly.

We hope that our responses have adequately addressed the reviewers’ concerns.

Reviewer’s comment 1
Thanks for your detailed revision. Your clarification and supplementation improve the quality of the manuscript. However, we could not figure out the beneficial effect of personalized lifestyle intervention on renal function from the study, which might relate to limited case numbers and different definitions of chronic kidney disease.

Response.  

We would like to appreciate the reviewer for his/her fruitful comment except one important issue regarding the significance of our study.

As the reviewer mentioned, we agree that it appears tough to definitively affirm the favorable effect of the home-visit lifestyle guidance on renal function, described in the current study.  This may be confirmed that little evidence has been published.  Therefore, we deeply considered and discussed the significance of our data again.

In general, people have their own lifestyle, so it is hard to make up the personalized lifestyle modification textbook for the specific person.  In this point, home-visit lifestyle guidance can provide more personalized guidance than group guidance.  This is a strong point of our study.  However, we also understand the several limitations of our study.  As we described in the manuscript, our study was not a randomized control study but an observational study with several unmeasurable confounding factors and without the control groups to compare with the intervention group.  In addition, we registered a small number of participants in this study because it is a method of visiting each person individually.  The net effect of the intervention could not be displayed, since we could not compare the subjects in this study with the non-intervention population.

From the result of our study, compared to the subjects who received the medical treatment, the subjects who did not receive medical treatment also tended to maintain renal function and improve urinary protein in the similar fashion.

First, the eGFR of the subjects with medical treatment decreased from 47.0 to 45.2 mL/min/1.73m2, thus the decline of eGFR was calculated as 1.8 mL/min/1.73 m2 in 4 years (means 0.45 mL/min/1.73m2/year).  On the other hand, the eGFR subjects without medical treatment decreased from 52.9 to 51.3 mL/min/1.73 m2, thus the decline of eGFR was calculated as 1.6 mL/min/1.73m2 in 4 years (means 0.4 mL/min/1.73m2/year).  The difference in the decline of eGFR between the groups was quite a small.  In the subjects without medical treatment, it is likely that their renal functions were preserved only by the lifestyle guidance.  In addition, reference to the cohort of the Japanese population with CKD, the average annual decline of renal function was 1.93 mL/min/1.73m2/year in CKD stage G3a and 2.06 mL/min/1.73m2/year in CKD stage G3b in the CKD-JAC study, and 1.9 mL/min/1.73 m2/year in the CKD stage 3 who received lifestyle guidance in the FROM-J study.  Although it is not appropriate to compare with other study subjects, our study subjects clearly demonstrate less decline in renal function, despite the lack of medical care for most participants.  What differ from other studies are regular contacts and personalized advices from the public health nurses.  Therefore, we believe that it is highly likely that our interventions had a significantly preferable impact on renal function.  

Second, regarding the definition of CKD, we described the definition in the text that “CKD is defined in examinees as eGFR less than 60 mL/min/1.73m2 or qualitative analysis of urinary protein 1+ or greater at the specific health check-up”.  In our study, to focus on higher-risk populations, we defined the subjects who had eGFR < 50 mL/min/1.73m2 and / or eGFR 50-90 mL/min/1.73m2 with qualitative analysis of urinary protein 1+ or greater in this study as home-visit guidance target.  In this point, all subjects in our study fits the original CKD definition.  For confirmation, we reviewed the data in detail again and then no participants significantly deviated from the CKD criteria during the 4-year follow-up.  Therefore, the reviewers’ concerns regarding the different definition of CKD seem less likely to significantly affect the results of the study.

The above contents are already described in detail in discussion and limitation paragraph of the manuscript.  We hope that our detailed reconsiderations above are properly conveyed to the reviewer.